# β-D-Glucose-Reduced Silver Nanoparticles Remodel the Tumor Microenvironment in a Murine Model of Triple-Negative Breast Cancer

**DOI:** 10.3390/ijms25158432

**Published:** 2024-08-02

**Authors:** Pedro Félix-Piña, Moisés Armides Franco Molina, Paola Leonor García Coronado, Heriberto Prado-Garcia, Diana Ginette Zarate-Triviño, Beatriz Elena Castro-Valenzuela, Kenia Arisbe Moreno-Amador, Ashanti Concepción Uscanga Palomeque, Cristina Rodríguez Padilla

**Affiliations:** 1Laboratorio de Inmunología y Virología, Facultad de Ciencias Biológicas, Universidad Autónoma de Nuevo León, San Nicolás de los Garza 66455, Mexico; dro_lix_09@hotmail.com (P.F.-P.); paolagarciacoronado@gmail.com (P.L.G.C.); dianazt@gmail.com (D.G.Z.-T.); bety.castro89@gmail.com (B.E.C.-V.); ashanti.uscanga@gmail.com (A.C.U.P.); crrodrig07@gmail.com (C.R.P.); 2Laboratorio de Onco-Inmunobiologia, Departamento de Enfermedades Crónico-Degenerativas, Instituto Nacional de Enfermedades Respiratorias Ismael Cosio Villegas, Mexico City 14080, Mexico; hpradog@yahoo.com

**Keywords:** breast cancer, silver nanoparticles, beta-D-glucose, microenvironment, PCNA, caspase

## Abstract

Breast cancer is the most diagnosed type of cancer worldwide and the second cause of death in women. Triple-negative breast cancer (TNBC) is the most aggressive, and due to the lack of specific targets, it is considered the most challenging subtype to treat and the subtype with the worst prognosis. The present study aims to determine the antitumor effect of beta-D-glucose-reduced silver nanoparticles (AgNPs-G) in a murine model of TNBC, as well as to study its effect on the tumor microenvironment. In an airbag model with 4T1 tumor cell implantation, the administration of AgNPs-G or doxorubicin showed antitumoral activity. Using immunohistochemistry it was demonstrated that treatment with AgNPs-G decreased the expression of PCNA, IDO, and GAL-3 and increased the expression of Caspase-3. In the tumor microenvironment, the treatment increased the percentage of memory T cells and innate effector cells and decreased CD4+ cells and regulatory T cells. There was also an increase in the levels of TNF-α, IFN-γ, and IL-6, while TNF-α was increased in serum. In conclusion, we suggest that AgNPs-G treatment has an antitumor effect that is demonstrated by its ability to remodel the tumor microenvironment in mice with TNBC.

## 1. Introduction

Breast cancer (BC) is the most commonly diagnosed neoplasm and the second cause of death in women in the United States [1]. In 2024, it is projected that there will be 2,001,140 new cancer cases and 611,720 cancer-related deaths in the United States. Triple-negative breast cancer (TNBC) is the most aggressive and is characterized by the absence of three specific receptors: the estrogen receptor (ER), the progesterone receptor (PR), and the human epidermal growth factor receptor 2 (HER2). The lack of specific targets makes TNBC the most challenging subtype to treat, and it has the worst prognosis [2]. Currently, chemotherapy, radiotherapy, and monoclonal antibodies are the treatment options [3].

TNBC is characterized by a tumor microenvironment (TME) that differs from other BC subtypes [4,5]. This complex environment is composed of diverse elements, such as stromal cells, blood vessels, immune cells, extracellular matrix (ECM) components, and soluble factors [6]. The dynamic interaction between cancer cells and the TME plays a crucial role in tumorigenesis [7]. Recent studies indicate the participation of stromal cells (which include fibroblasts and cells of the immune system) in tumor progression by promoting cell proliferation, neovascularization, and evasion of immune control [8].

Numerous studies have shown that the TME significantly influences the therapeutic response by conferring resistance to chemotherapy [9,10]. This resistance can be induced through soluble factors secreted by cancer and stromal cells. Adhesion of cancer cells to cancer-associated fibroblasts (CAFs) or ECM elements could also be another attenuating factor [11,12]. However, the complexity and heterogeneity of the TME during the evolution of BC and after treatment still need to be clarified.

The complex structure of the TME imposes great challenges not only for the understanding of its components but also for the development of effective therapies. To combat this problem, new therapeutic approaches have been proposed, directed at both the tumor cells and multiple components of the TME, including immune cells [13,14].

In this context, studies have shown that silver nanoparticles (AgNPs) have antitumor effects by inducing apoptosis in cancer cells and interfering with key processes in proliferation. Furthermore, their antitumor effect in vivo was greater in immunocompetent mice than in their immunodeficient counterparts, suggesting that administration of AgNPs activates the antitumor immunity of the TME [15,16].

Previously, our research group synthesized beta-D-glucose-reduced silver nanoparticles (AgNPs-G) and demonstrated that BC cells exhibit sensitivity to treatment with AgNPs-G, including cells with a TNBC subtype. Furthermore, treatment with AgNPs-G affects the cell cycle and partially induces an immunogenic cell death mechanism in immunocompetent mice through the release of damage-associated molecular patterns (DAMPs) [17].

Based on all the above, the present study aims to determine the antitumor effect of AgNPs-G in a murine model of TNBC as well as to study its effect on the tumor microenvironment.

## 2. Results

### 2.1. Treatment with AgNPs-G Decreased Tumor Volume and Prolonged Survival of Mice Bearing 4T1 Tumor

Antitumor activity was evaluated using an airbag model with BC 4T1 tumor cell implantation. As can be seen in Figure 1A,B, the volume and weight of the tumor were significantly (*p* ≤ 0.05) lower in the groups treated with AgNPs-G and doxorubicin than in the control group. In turn, both groups demonstrated increased survival time of the tumor-bearing mice compared to the control group (Figure 1C).

### 2.2. Treatment with AgNPs-G Decreased the Expression of PCNA, IDO, and Gal-3 and Increased Caspase-3

Treatment with AgNPs-G significantly decreased (*p* ≤ 0.05) expression of the proliferating cell nuclear antigen (PCNA) (37.4%), indoleamine 2,3-dioxygenase (IDO) (53.8%), and galectin-3 (GAL-3) (41.9%) markers compared to the control group (Figure 2). On the other hand, AgNPs-G increased the expression of Caspase-3 (28.9%).

### 2.3. Treatment with AgNPs-G Increased the Percentage of Cytotoxic T Cells, Memory Cells, and Innate Effector Cells in Tumor Tissue

Once the antitumor effect of AgNPs-G was determined, we decided to evaluate the leukocyte populations found in the TME. As can be seen in Figure 3, the treatment with AgNPs-G significantly (*p* ≤ 0.05) increased the percentage of CD8+ cells (59.3%) compared to the control group (18.6%).

In turn, AgNPs-G increased the percentage of memory T cells (34.1%) and innate effector cells (18.7) in comparison to the control group (9.7% and 14.8%, respectively). Treatment with AgNPs-G also decreased the percentage of CD4+ cells (17.6%) and regulatory T cells (3.2%) compared to the control group (79.7% and 21.3%, respectively).

### 2.4. Treatment with AgNPs-G Increased the Levels of TNF-α, IFN-γ, and IL-6 in Tumor Tissue

At the tumor microenvironment level, treatments with AgNPs-G and doxorubicin significantly increased (*p* ≤ 0.05) the levels of TNF-α, IFN-γ, and IL-6 compared to the control group (Figure 4).

On the other hand, both treatments significantly decreased (*p* ≤ 0.05) the levels of IL-2, IL-4, and IL-10 compared to the control group. In the case of IL-17A, only treatment with doxorubicin increased levels significantly (*p* ≤ 0.05).

### 2.5. Treatment with AgNPs-G Increased Serum Levels of TNF-α

Once the levels of cytokines in tumor tissue were determined, we decided to quantify the serum levels. As can be seen in Figure 5, treatment with AgNPs-G only significantly increased (*p* ≤ 0.05) the TNF-α levels in comparison to the control group.

On the other hand, treatment with AgNPs-G significantly decreased (*p* ≤ 0.05) the serum levels of IL-2, IL-4, and IL-10.

### 2.6. Treatment with AgNPs-G Did Not Cause Significant Changes in Hematological Parameters

As can be seen in Table 1, untreated (control) and mice treated with AgNPs-G did not show significant differences in hematological parameters. However, both groups had a low red blood cell count, and the hematocrit and hemoglobin percentages were below the standardized normal range.

### 2.7. Treatment with AgNPs-G Increased the Levels of Liver Enzymes in Mice Bearing 4T1 Tumor

As can be seen in Table 2, treatment with AgNPs-G increased the levels of aspartate aminotransferase (AST) (58.3 ± 6.33 IU/L), alanine aminotransferase (ALT) (49.21 ± 8.2 U/L), and alkaline phosphatase (ALP) (147.23 ± 33.02 U/L) significantly (*p* ≤ 0.05) compared to the control group (26.23 ± 8.23 IU/L, 32.47 ± 6.78 U/L, and 82.08 ± 12.76 U/L, respectively). On the other hand, a decrease in total protein (5.3 ± 1.04 g/dL) was observed in the group treated with AgNPs-G compared to the control group (6.5 ± 0.42 g/dL).

### 2.8. Treatment with AgNPs-G Generated Histomorphological Changes in the Liver, Kidney, Lung, Heart, or Brain

The histological analysis ruled out the presence of perceptible lesions, necrotic areas, or signs of erosion in the main organs (liver, kidney, lung, heart, and brain) in any experimental group (Figure 6). Likewise, the presence of lymphocytic infiltrates or signs of malignancy was ruled out.

## 3. Discussion

In the present study, we investigated the therapeutic potential of AgNPs-G in the treatment of BC, as well as its effect on the TME. Our results indicate that treatment with AgNPs-G delays tumor growth and increases survival in mice. Although this effect can be attributed to several mechanisms, previous research by our team has shown that AgNPs-G interferes with the cell cycle of multiple BC cell lines and leads to death associated with the release of DAMPs [17]. These findings are consistent with previous studies that have demonstrated the in vivo therapeutic potential of AgNPs in the treatment of TNBC [18,19]. These mechanisms may have been facilitated by incorporating beta-D-glucose in synthesizing our particles, which could be present on their surface, thereby enhancing their interaction with cancer cells. However, the exact location of beta-D-glucose in our particles remains unknown.

The mechanism of action of doxorubicin is based on the damage it causes to cellular DNA and its ability to produce free radicals [20]. These molecular mechanisms also arise in treatments based on other AgNPs [21]. For this, we can hypothesize that perhaps our nanoparticles have a mechanism of action similar to this chemotherapeutic agent used as a control. However, it is crucial to perform clinical trials that allow us to determine with certainty that our AgNPs and doxorubicin have similar mechanisms of action.

On the other hand, our results show a decrease in the expression of the proliferation marker PCNA and the molecules IDO and GAL-3 in the group treated with AgNPs-G compared to the control. The reduction in PCNA expression correlates with the reduction in tumor volume, suggesting that treatment with AgNPs-G has a direct effect on the divisional capacity of tumor cells, which has also been described in studies using a small molecule (PCNA-I1S) that can directly bind to PCNA, stabilize the trimer structure, and reduce chromatin-associated PCNA, thereby selectively inhibiting tumor cell growth and inducing apoptosis [22].

Furthermore, the decrease in the expression of IDO and GAL-3 is relevant, since these proteins are associated with the promotion of immunosuppressive tolerance and tumor progression [23,24]. On the other hand, the increase in caspase-3 expression suggests that AgNPs-G could promote apoptotic pathways. Similar data were reported in the 5637 bladder cancer cell line in which the AgNPs induced excessive ROS formation, up-regulation of Bax/Bcl-2 expression, and caspase 3 and 7 activation [25]. Subsequently, we evaluated the implication of treatment with AgNPs-G on leukocyte populations within the TME. Our results indicate that treatment with AgNPs-G increases the percentage of CD8+ T cells. This finding is important due to the central role of CD8+ T cells in the direct elimination of tumor cells, indicating an improvement in the ability of the immune system to recognize and effectively attack cancer cells [26]. Furthermore, we found an increase in the percentage of memory cells and innate effector cells, which could be related to the previously reported release of DAMPs and the induction of set-specific T cells able to infiltrate and eliminate the tumor [27]. Treatment with AgNPs-G decreased the percentage of CD4+ cells and regulatory T cells in tumor tissue. Recently a clinical trial on 5177 breast cancer patient samples demonstrated that low Treg abundance was associated with a pathological complete response after neoadjuvant chemotherapy in TNBC [28]. Possibly, a concomitant treatment with AgNps can be of utility to obtain a better treatment response in cancer patients. However, some limitations can be related to the ethical approval, route of administration, and doses in human patients.

Our findings reveal a significant increase in the levels of TNF-α, IFN-γ, and IL-6 in tumor tissue following treatment with AgNPs-G. These proinflammatory cytokines are known to play a key role in the activation and regulation of antitumor immune responses. TNF-α is involved in inflammation and has a dual function in cancer as an inductor of metastasis or in tumor elimination [29]. However, TNF-α is also involved in host defense, regulation of cytotoxic activity against cells and tumors, activation of T lymphocytes and NK cells, stimulation of monocytes to generate cytokine production, and activation of endothelium and adipocyte differentiation [30]. In the case of IFN-γ, multiple studies have shown that inhibitors of CTLA-4 and PD-1, as well as other immune checkpoint blockade therapies, increase IFN-γ production, which subsequently results in the elimination of cancer cells. Recently, it has been confirmed that resistance to immunotherapy is associated with defects in IFN-γ signaling. These findings suggest that the efficacy of cancer immunotherapy is, at least in part, due to an increase in IFN-γ expression [31]. Similar data indicate an increase of IL-6 after treatment with 20 or 200 nm of AgNPs on MDA-MB-436 cells [32]. The application of IL-6 on breast cancer cells inhibits proliferation in estrogen receptor-positive cells, while high circulating IL-6 levels are correlated with a poor prognosis in breast cancer patients, indicating discrepancies and the distinct roles of this cytokine [33].

On the other hand, AgNPs-G treatment significantly decreased the levels of IL-2, IL-4, and IL-10 in tumor tissue compared to the control group. These cytokines are associated with immunosuppressive and regulatory functions, so their decrease could promote a better response to cancer treatment, especially IL-10, given its key role in tumorigenesis and its negative correlation with disease-free survival in certain cancers [34]. Thereby, the results of our study suggest in part that cytokines induced by AgNps-G treatment can be involved in modifying the tumor microenvironment and induce an immune response to reduce tumor mass. However, some limitations, such as the lack of mutant mice specific to each cytokine to clarify its participation in the tumor microenvironment, restrict the reach of our results.

Hematological parameters could indicate adverse effects of the compounds on blood constituents [35]. The treatment with AgNPs-G increased the hematological values related to AST, ALT, and ALP in the mice treated with AgNPs-G compared to the control group. The increase in these hepatic enzymes is indicative of damage in certain organs that correlates with histological data. Previous studies reported a high incidence of anemia in cancer patients [36] and a correlation with poor therapy outcomes and prognosis in breast cancer patients [37,38]. Although hematocrit, hemoglobin, and red blood cell levels were below the reference levels, such findings appear to be independent of treatment, as there were no significant differences in hematocrit and hemoglobin among the experimental groups. 

In conclusion, we suggest that AgNPs-G treatment has an antitumor effect that is enhanced by their ability to remodel the tumor microenvironment in mice with TNBC. However, additional studies are needed to fully understand the exact mechanisms of action of AgNPs-G, optimize treatment strategies, and evaluate their long-term safety in preclinical models and, eventually, in clinical trials.

## 4. Materials and Methods

### 4.1. Cell Line

The 4T1 murine breast cancer cell line (ATCC^®^ CCL-2539™) was obtained from ATCC (American Type Culture Collection, Manassas, VA, USA). The cells were cultured in Dulbecco’s modified Eagle’s medium (DMEM) (GIBCO^®^, Thermo Scientific, Waltham, MA, USA) supplemented with 10% fetal bovine serum (FBS) (GIBCO^®^, Thermo Scientific, Waltham, MA, USA), and antibiotic- 1% antifungal (penicillin, streptomycin, and amphotericin B) (Sigma, St. Louis MO, USA). The cell line was maintained under specific conditions at a temperature of 37 °C, relative humidity ≈85%, and an atmosphere of 95% air and 5% CO_2_.

### 4.2. Animals

Six- to eight-week-old female BALB/c mice were used. They were maintained under bioterium conditions (temperature of 25 °C, relative humidity of ≈55%, and 12 h light/dark cycles). All mice received food and water ad libitum. All protocols were carried out following the official Mexican animal welfare standard NOM-062-ZOO-1999 and were previously approved by the Animal Bioethics Committee of the Facultad de Ciencias Biologicas of the Universidad Autónoma de Nuevo León (San Nicolas de los Garza, NL, Mexico).

### 4.3. Beta-D-Glucose-Reduced Silver Nanoparticles (AgNPs-G)

The synthesis of AgNPs-G was carried out following the methodology previously described by our work team [17], which is summarized as follows: 10 mL of a 0.3 M aqueous solution of β-D-glucose was placed in a beaker at 120 °C for 5 min. Next, 100 μL of 2.5 mM AgNO_3_ solution and 10 μL of 0.1 M NaOH solution were added until a yellow color change characteristic of the formation of AgNPs-G was observed. These AgNPs were previously characterized and have an average size of 5.991 nm, quasi-spherical morphology, and a −7.48 mV zeta potential value.

### 4.4. Airbag Model, Tumor Implantation, and Treatment Administration

For tumor implantation, an airbag model was used, which is a technique that allows precise evaluation of the tumor and its environment through the creation of a confined space [24]. The procedure is detailed below. Six days before the inoculation of the tumor cells, the mouse’s back was shaved, and 5 mL of sterile air was injected subcutaneously. Three days later, 3 mL of sterile air was injected again.

Viable 4T1 cells (0.5 × 10^6^) were injected into the air pocket. Nine days later, having reached a tumor size of approximately 100 mm^3^, the mice were randomly divided into three experimental groups (n = 6): (1) control group, no treatment, (2) doxorubicin group, single dose of doxorubicin (Doxolem^®^, Teva Pharmaceuticals, Mexico) (10 mg/kg, peritumoral route), and (3) AgNPs-G group, a daily dose for 7 days of AgNPs-G (1.08 mg/kg, peritumoral route).

Tumor volume was measured daily using a vernier caliper until the day of sacrifice (30 days after inoculation of 4T1 cells) and was defined using the following equation [39].
Tumor volume: 4/3 × π × L/2 × W/2 × h/2
where L corresponds to the longest side, W to the shortest side, and h to the height of the tumor.

The tumors once removed (postmortem) were weighed using a TE241S laboratory analytical balance (Sartorius, Goettingen, Germany). For the animal survival assay, five additional mice per group were maintained for 60 days post-inoculation with 4T1 cells. Mice were sacrificed if the tumor volume reached 2000 mm^3^.

### 4.5. Immunohistochemistry

The tumors were dissected and fixed in 10% neutral formalin (pH 7.2) for 24 h for subsequent paraffin embedding. Tumor sections of 3 to 5 μm were cut, deparaffinized, and hydrated in xylol-alcohol. Subsequently, the samples were incubated with sodium citrate buffer (10 mM sodium citrate, 0.05% Tween 20, pH 6.0) for 30 min at a temperature of 60 °C. Samples were blocked with normal horse serum (Vector Laboratories, Newark, CA, USA). Samples were incubated separately with the primary antibodies: anti-CTLA-4 sc-376016 (Santa Cruz Biotechnology, Dallas, Texas, USA), anti-IDO sc-137012 (Santa Cruz Biotechnology, CA, USA), anti-Gal-3 sc-32790 (Santa Cruz Biotechnology, CA, USA), anti-Caspase-3 ab214430 (Abcam Inc., Walthman, MA, USA), and anti-PCNA sc-528093 (Santa Cruz Biotechnology, Dallas, Texas, USA). All antibodies were incubated at 4 °C for 24 h (1:1000 dilution). A biotinylated pan-specific universal antibody (Vector Laboratories, Newark, CA, USA) was used as a secondary antibody. Samples were counterstained with hematoxylin (Sigma Aldrich, St. Louis, MO, USA) and dehydrated in a xylol-alcohol gradient for inclusion in a mounting medium (Entellan^®^, Merck Millipore, Darmstadt, Germany).

Micrographs were captured using a Zeiss Imager Z1 microscope (Zeiss, Germany) coupled with a Ve-LX1000 camera (VELAB, Tlalpan, CDMX, Mexico). The images were processed using Velabview software version 2.0 (VELAB, Tlalpan, CDMX, Mexico). Positive staining for DAB was evidenced by the presence of brown cells and was quantified using Fiji image processing software (ImageJ version 2.0).

### 4.6. Determination of Cytokines

Cytokine levels were analyzed from the serum and tumor tissue obtained 30 days post inoculation of the tumor cells. For serum collection, peripheral blood was centrifuged at 3600 rpm for 10 min at 4 °C. On the other hand, the tumors were cut and incubated at 4 °C with lysis buffer (150 mM sodium chloride, 1% Triton 100-X, 50 mM Tris, Halt ™ protease inhibitor cocktail, and pH 8.0) for 30 min under gentle stirring. Subsequently, the lysed tumor sections were centrifuged at 12,000 rpm for 20 min at 4 °C, and the supernatants were collected for subsequent analysis.

Cytokine levels were quantified using the BD Cytometric Bead Array Mouse Inflammation Kit (BD Horizon, Franklin Lakes, NJ, USA) following the manufacturer’s instructions. Data acquisition was performed using a BD Accuri ^TM^ C6 flow cytometer (BD Horizon, Franklin Lakes, NJ, USA). Finally, the data obtained were analyzed using CFlow plus software version 1.0.264.15 (BD Biosciences, Milpitas, CA, USA).

### 4.7. Leukocyte Isolation

The tumors were cut and incubated with gentle shaking with 0.1 mg/mL Liberase ^TM^ TL solution (Roche, Mannheim, Germany) for 30 min at 37 °C. Subsequently, leukocytes were obtained by density gradient centrifugation using Ficoll-Histopaque 1077 (Sigma, St Louis, MO, USA) following the manufacturer’s instructions.

### 4.8. Leukocyte Immunophenotyping

Leukocytes were labeled with anti-mouse antibodies: anti-CD3 FITC 555274, anti—CD8 PE 553033, anti-CD4 APC 553051, anti-CD44 PE 51-9007324, anti-CD62 L APC 5-9007326, anti-CD16/CD32 APC 558636, anti-CD25 PE 12-0251-81, and anti-FOXP3 PE-Cy5 15-5773-80A; all manufactured by BD Biosciences, CA, USA.

For FOXP3 labeling only, cells were prefixed with formaldehyde (4% *v*/*v* in PBS) for 1 min, permeabilized with 90% methanol for 30 min in an ice water bath, washed, and resuspended in PBS.

All samples with their respective antibodies were incubated in dark conditions for 30 min at room temperature, washed with albumin (0.5% *w*/*v* in PBS), and resuspended in PBS for subsequent analysis. Data acquisition was performed using a BD Accuri ^TM^ C6 flow cytometer (BD Horizon, Franklin Lakes, CA, USA).

### 4.9. Blood and Serum Analysis

Blood was collected on day 30 post inoculation of the tumor cells for each experimental group in microtainer tubes with K _2_ EDTA (blood), lithium heparin, and gel (serum). Hematological parameters were analyzed using a BC2800 hematological analyzer (Mindray, Shenzhen, China), while biochemical parameters were analyzed using a Skyla VB1 automated chemistry analyzer (Skyla, Hsinchu, Taiwan).

### 4.10. Hematoxylin-Eosin Staining

The brain, lungs, heart, liver, and kidneys were removed (postmortem) from the mice in each experimental group and fixed in formalin (10% in PBS and pH 7.2) for 24 h for subsequent paraffin embedding. Tumors were sectioned into 3 to 5 μm sections, deparaffinized, and hydrated in a xylol-alcohol gradient. The tissues were then stained with Mayer’s hematoxylin solution for 3 min and counterstained with Eosin-phloxin B solution for 30 s. Finally, the samples were dehydrated in an xylol-alcohol gradient for inclusion in a mounting medium (Entellan^®^, Merck Millipore, Darmstadt, Germany).

### 4.11. Statistical Analysis

All experiments were carried out in triplicate with an analysis of variance (ANOVA)-type experimental study followed by Tukey’s post hoc test using GraphPad Prism software version 8.0.2 (San Diego CA, USA). *p* values were considered significant as follows: *p* < 0.05 (*).

## Figures and Tables

**Figure 1 ijms-25-08432-f001:**
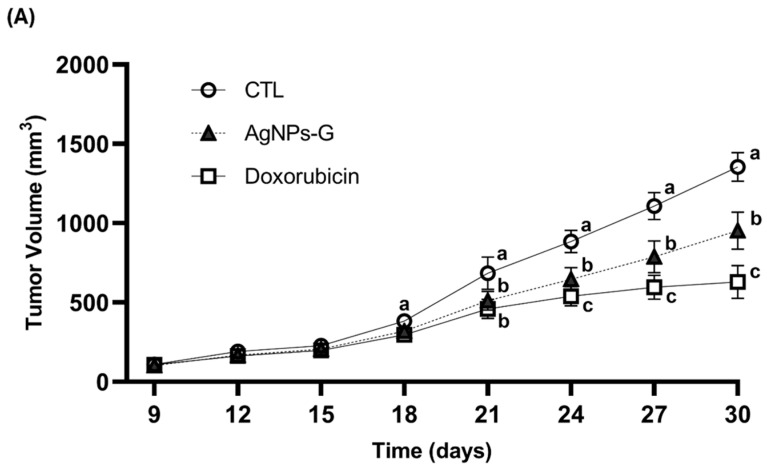
Antitumor activity of AgNPs-G in vivo. (**A**) The tumor volume (mm^3^) was measured daily from inoculation until the day of sacrifice (day 30). (**B**) The tumors once removed (postmortem) were weighed. (**C**) For the animal survival assay, five additional mice per group were maintained for 60 days post inoculation. The survival of the mice was represented by the Kaplan–Meier plot. Statistical analysis was performed by ANOVA using Tukey’s test. Letters (a–c) show significant differences *p* ≤ 0.05 (*) between treatments.

**Figure 2 ijms-25-08432-f002:**
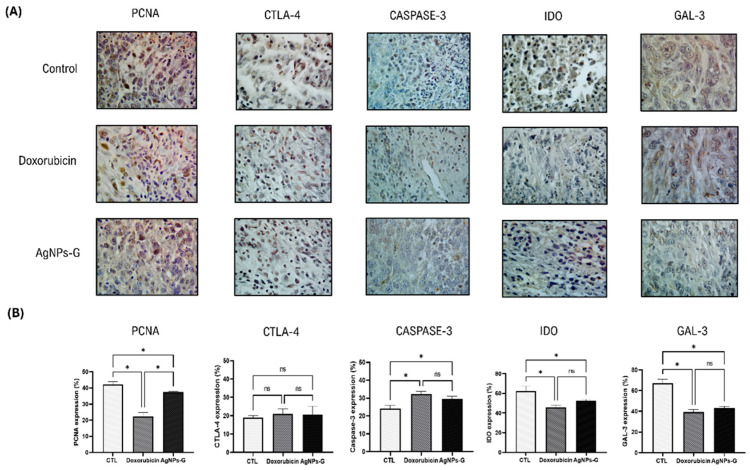
Immunohistochemical staining and percentage of expression of PCNA, CTLA-4, caspase-3, IDO, and Gal-3 in tumor tissue. Thirty days after inoculation with viable BC 4T1 cells, tumors were removed (postmortem) from untreated mice (control) and treated with: AgNPs-G or doxorubicin. Tumors were fixed, paraffin-embedded, and cut into 3 to 5 μm sections. Subsequently, immunohistochemical staining was performed. (**A**) Representative immunohistochemistry micrographs (magnification 20×). (**B**) Bar graphs represent the mean percentage expression obtained by color deconvolution analysis. Statistical analyses were performed by ANOVA using Tukey’s post hoc test; *p* ≤ 0.05 (*) and ns (not significant).

**Figure 3 ijms-25-08432-f003:**
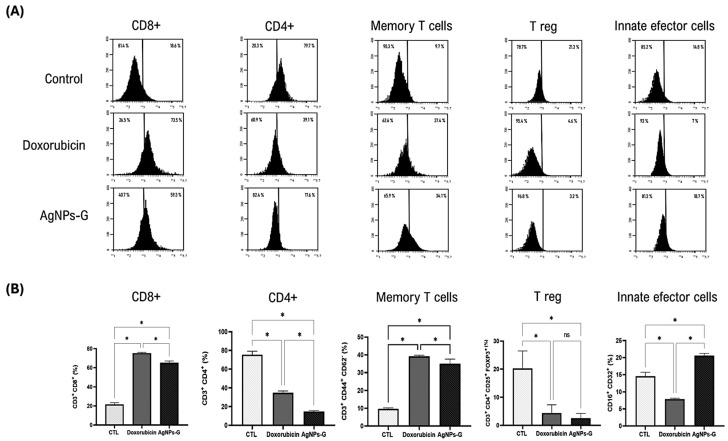
Leukocyte subpopulations of tumor tissue. Thirty days after inoculation with viable BC 4T1 cells, tumors were removed (postmortem) from untreated mice (control) or mice treated with AgNPs-G or doxorubicin. Tumors were cut and incubated in Liberase^TM^ TL solution, and leukocytes were isolated by density gradient centrifugation using Ficoll-Histopaque. (**A**) Representative histograms of the leukocyte subpopulations CD8+, CD4+, memory T cells, regulatory T cells, and innate effector cells determined by flow cytometry; the Y-axis represents the cell count. (**B**) Bar graphs represent the average cell population percentage. Statistical analyses were performed by ANOVA using Tukey’s post hoc test; *p* ≤ 0.05 (*) and ns (not significant).

**Figure 4 ijms-25-08432-f004:**
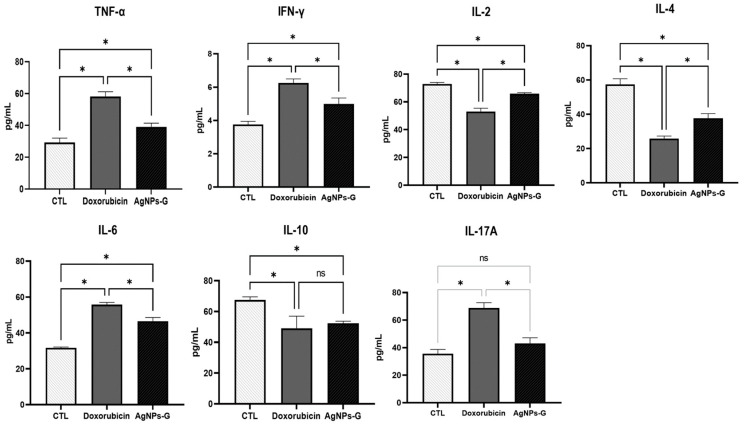
Treatment with AgNPs-G increased the levels of TNF-α, IFN-γ, and IL-6 in tumor tissue. Thirty days after inoculation with viable BC 4T1 cells, tumors were removed (postmortem) from untreated mice (control) and mice treated with AgNPs-G or doxorubicin. Tumors were cut and incubated in lysis buffer for 30 min with gentle shaking. Subsequently, the lysed tumor sections were centrifuged, and the supernatants were collected for subsequent analysis. Bar graphs represent cytokine levels determined by flow cytometry analysis with the BD Cytometric Bead Array Mouse Inflammation Kit. Statistical analyses were performed by ANOVA using Tukey’s post hoc test; *p* ≤ 0.05 (*) and ns (not significant).

**Figure 5 ijms-25-08432-f005:**
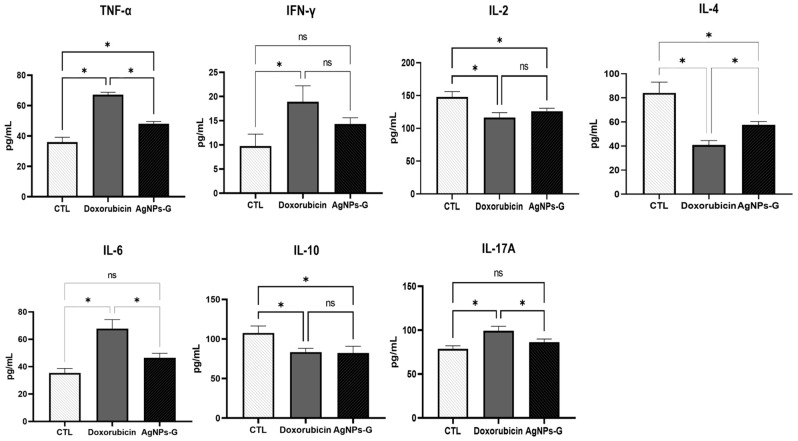
Treatment with AgNPs-G increased serum levels of TNF-α. Thirty days after inoculation with viable BC 4T1 cells, serum was collected from untreated mice (control) and mice treated with AgNPs-G or doxorubicin. Bar graphs represent cytokine levels determined by flow cytometry analysis with the BD Cytometric Bead Array Mouse Inflammation Kit. Statistical analyses were performed by ANOVA using Tukey’s post hoc test; *p* ≤ 0.05 (*) and ns (not significant).

**Figure 6 ijms-25-08432-f006:**
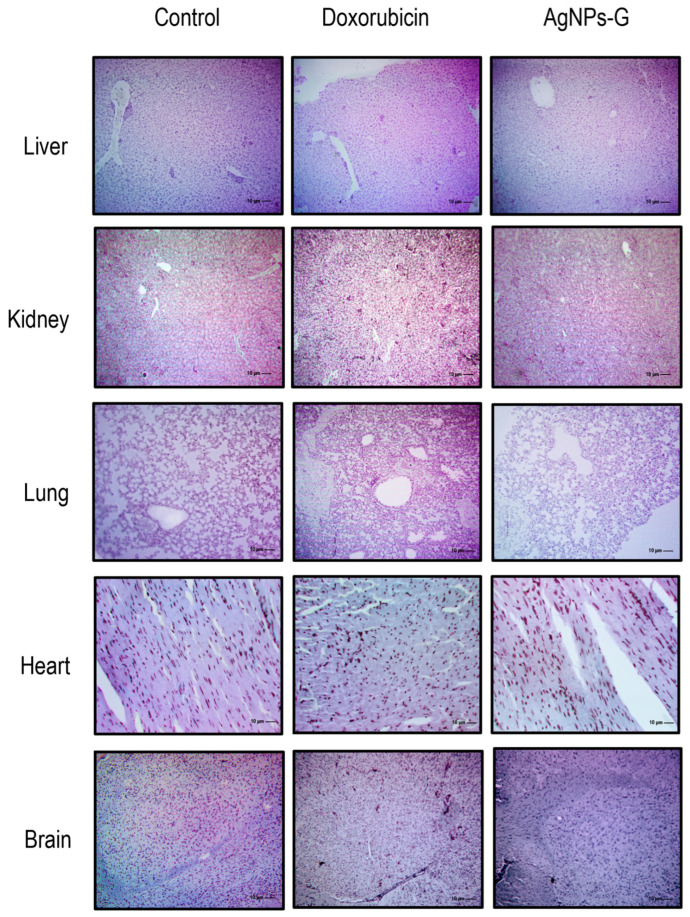
Micrographs of the liver, kidney, lung, heart, and brain. Thirty days after inoculation with viable BC 4T1 cells, organs from untreated mice (control) and mice treated with AgNPs-G or doxorubicin were removed (postmortem) (n = 6). All organs were fixed in formalin for 24 h until embedding in paraffin. Tissue sections of 3 to 5 μm were stained with hematoxylin-eosin solution. Micrographs were taken using a Zeiss Imager Z1 microscope was coupled with a Ve-LX1000 camera.

**Table 1 ijms-25-08432-t001:** Hematological parameters of Balb/C mice bearing 4T1 tumors.

	Control	Doxorubicin	AgNPs-G	Normal Range
Neutrophils (%)	41.62 ± 9.04	49.10 ± 14.23	44.2 ± 10.28	35.0–75.0
Lymphocytes (%)	46.72 ± 6.22	42.5 ± 5.19	41.87 ± 4.61	25.0–50.0
Monocytes (%)	6.78 ± 1.23	6.23 ± 0.75	6.54 ± 2.01	2.0–10.0
Eosinophils (%)	3.12 ± 0.68	4.12 ± 1.08	3.56 ± 0.75	1.0–4.0
Total white blood cell count (10^9^/L)	5.21 ± 1.26 ^a^	6.98 ± 0.48 ^b^	5.68 ± 0.72 ^a^	4.0–10.0
Red Blood Cells (10^12^/L)	7.57 ± 0.52 ^a^	6.89 ± 0.23 ^b^	7.11 ± 0.09 ^a^	8.0–15.0
Hematocrit (%)	39.32 ± 4.12	38.77 ± 3.12	37.54 ± 3.31	40.0–54.0
Hemoglobin (g/dL)	13.19 ± 0.82	13.32 ± 1.02	12.44 ± 0.87	14.0–17.5

Thirty days after inoculation with viable BC 4T1 cells, blood was collected from untreated mice (control) and mice treated with AgNPs-G or doxorubicin. Hematological parameters were analyzed with the Sysmex XS 1000™ Hematology Analyzer. Values are presented as mean ± SD (*p* ≤ 0.05). Statistical significance was performed by ANOVA using Tukey’s post hoc test. Letters (a and b) show a significant difference (*p* ≤ 0.05) between treatments.

**Table 2 ijms-25-08432-t002:** Biochemical parameters of tumor-bearing mice.

	Control	Doxorubicin	AgNPs-G	Normal Range
Total bilirubin (mg/dL)	0.22 ± 0.06	0.25 ± 0.05	0.32 ± 0.08	0.0–1.40
Direct bilirubin (mg/dL)	0.07 ± 0.008	0.08 ± 0.01	0.11 ± 0.03	0.0–0.50
Indirect bilirubin (mg/dL)	0.04 ± 0.01	0.06 ± 0.02	0.06 ± 0.01	0.0–0.90
Total protein (g/dL)	6.5 ± 0.42 ^a^	5.58 ± 1.23 ^b^	5.3 ± 1.04 ^b^	6.0–8.10
Albumin	3.72 ± 0.52	3.12 ± 0.61	3.4 ± 0.41	3.0–5.20
Globulin	2.88 ± 0.58	2.38 ± 0.43	2.23 ± 0.38	1.5–3.30
Aspartate aminotransferase (UI/L)	26.23 ± 8.23 ^a^	43.2 ± 12.1 ^b^	58.3 ± 6.33 ^c^	4.0–37.0
Alanine aminotransferase (U/L)	32.47 ± 6.78 ^a^	57.25 ± 6.5 ^b^	49.21 ± 8.21 ^b^	4.0–41.0
Alkaline Phosphatase (U/L)	82.08 ± 12.76 ^a^	135.8 ± 18.01 ^b^	147.23 ± 33.02 ^b^	40.0–129.0

Thirty days after inoculation with viable BC 4T1 cells, blood was collected from untreated mice (control) and treated with AgNPs-G or doxorubicin. Biochemical parameters were analyzed using the Skyla VP1™ Chemistry Analyzer. Values are presented as mean ± SD (*p* ≤ 0.05) (n = 6). Statistical analysis was performed by ANOVA using Tukey’s post hoc test. Letters (a–c) show a significant difference (*p* ≤ 0.05) between treatments.

## Data Availability

The data presented in this study are available on request from the corresponding author.

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
