# Peer review of "β-D-Glucose-Reduced Silver Nanoparticles Remodel the Tumor Microenvironment in a Murine Model of Triple-Negative Breast Cancer"

_ijms, 2024, doi:10.3390/ijms25158432_

Round 1

Reviewer 1 Report

Comments and Suggestions for Authors

The manuscript by Pina et al. entitled “β-D-Glucose – Reduced silver nanoparticles remodel the tumor microenvironment in a murine model of triple-negative breast cancer” describes the analysis of potential antitumor activity of AgNPs in murine model of breast cancer. Although the manuscript may be interesting for the readership of IJMS, in my opinion the major improvements are needed before it can be accepted for publication. The specific comments are given below:

1)      Page 1, lines 30-31: The authors stated that “ Breast cancer is the most commonly diagnosed neoplasm worldwide and is the leading cause of death in women.” The authors give reference to breastcancer.org website to support this statement. However, according to breastcancer.org: ” Breast cancer is the most common cancer diagnosed among U.S. women and the second leading cause of death from cancer among women after lung cancer, according to the American Cancer Society. It’s also the most common cancer in the world.” Authors should check and correct their sentence.

2)      The authors should discuss the fact that the observed effects of AgNPs are very similar to the effects of doxorubicin. Is it possible that their mechanisms of action are similar?

3)      Description of figure 4 given in paragraph 2.4 doesn’t match the results presented in the figure. Similarly, description of figure 5 doesn’t match the figure. It seems that the figures 4 and 5 have been switched. Please check and correct.

4)      For different analyses, different p values are regarded as statistically significant: p<0.033 (e.g. fig 4, 5) or p<0.05 (e.g. table 1, table 2). Please explain this inconsistency. Moreover, description of figures 4 and 5 says that differences with p<0.12 are not significant. This is rather uncommon approach. Please check and correct or explain.

5)      It should be discussed why all groups have red blood cells counts, hematocrit and hemoglobin below normal range (paragraph 2.6).

6)      Page 10, lines 205-206: The sentence is incomplete. Please correct.

7)      The characteristics of AgNPs should be given (e.g. what is their size?). Alternatively, reference to the publication where such characteristics can be found, should be given.

8)      Description of the experimental model is incomplete:

-          route of AgNPs administration is unknown;

-          AgNPs dose should be calculated an administrated in mg/kg body mass just like doxorubicin;

-          how were tumor dimensions measured?

-          how many animals was in each experimental group?

-          Page 13 line 380: “All experiments were carried out in triplicate” – what does it mean? Three animals per group? Three replicates of each measurement? Whole experiment was repeated three times? Please explain.

9)      Is it possible that as a result of using B-D-Glucose as a reducing agent, B-D-Glucose is present on the surface of AgNPs and is responsible (at least partially) for the observed effects?

10)   Did the authors exclude the presence of pyrogens in AgNPs solution used in the experiments?

Author Response

Reviewer 1

Dear reviewer we would like to thank you for your comments and attentive review, it greatly helped us improve the manuscript and correct mistakes that were overlooked.  Additionally, the English language was revised, and minor changes were made to the main manuscript.

Please find below the corrections made according to the comments.

1)      Page 1, lines 30-31: The authors stated that “ Breast cancer is the most commonly diagnosed neoplasm worldwide and is the leading cause of death in women.” The authors give reference to breastcancer.org website to support this statement. However, according to breastcancer.org: ” Breast cancer is the most common cancer diagnosed among U.S. women and the second leading cause of death from cancer among women after lung cancer, according to the American Cancer Society. It’s also the most common cancer in the world.” Authors should check and correct their sentence.

Thank you for your comment, we have corrected the wording of the sentence as follows:

Breast cancer (BC) is the most commonly diagnosed neoplasm and a leading cause of death in women in the United States [1].

2)      The authors should discuss the fact that the observed effects of AgNPs are very similar to the effects of doxorubicin. Is it possible that their mechanisms of action are similar?

The doxorubicin was included as positive control. A similar capacity to affect cellular viability, not mean a similar mechanism molecular of action. For answer your question correctly is necessary stablish a design experimental, at molecular level address to a specific target that affect doxorubicin. Furthermore, stablish clinical trial in dogs or humans to can conclude that AgNps and doxorubicin possesses similar molecular mechanism.

3)      Description of figure 4 given in paragraph 2.4 doesn’t match the results presented in the figure. Similarly, description of figure 5 doesn’t match the figure. It seems that the figures 4 and 5 have been switched. Please check and correct.

We apologize for overseeing the mistake, the figures were indeed switched during manuscript editing. We have arranged them, accordingly, please find the correct order in the manuscript.

4)      For different analyses, different p values are regarded as statistically significant: p<0.033 (e.g. fig 4, 5) or p<0.05 (e.g. table 1, table 2). Please explain this inconsistency. Moreover, description of figures 4 and 5 says that differences with p<0.12 are not significant. This is rather uncommon approach. Please check and correct or explain.

We appreciate your comment. We made our statistical analysis and graphs using the GraphPad software, the program uses de GrapgPad (GP) p value style in which the significance output is represented as ns 0.01234,  p < 0.0332 (*), p < 0.0021 (**), p < 0.0002 (***) and p < 0.0001 (****) after Tukey's post hoc test. However, this is only to represent the significance output, significance and confidence during statistical analysis are set accordingly at 0.05 (95% confidence interval). In the analyze differences significance values are obtained when the means evaluated statistically and this, are reflected in the graphic for a better explanation. 

Regarding tables the change of p value style, we considered the use of letters to indicate the differences among groups in the different parameters easier to understand in comparison with the * in graphs.

5)      It should be discussed why all groups have red blood cells counts, hematocrit and hemoglobin below normal range (paragraph 2.6).

Discussion of the results has been included, please find them in line 262-267 of the manuscript and as stated below:

Previous studies reported a high incidence of anemia in cancer patients [34] and a correlation with poor therapy outcomes and prognosis in breast cancer patients [35,36]. Although, hematocrit, hemoglobin, and red blood cell levels are below the reference levels such findings appear to be independent of treatment as there were no significant differences in hematocrit and hemoglobin among experimental groups.

6)      Page 10, lines 205-206: The sentence is incomplete. Please correct.

The sentence has been corrected, please find the revised version below:

in studies using a small molecule (PCNA-I1S) which can directly bind to PCNA, stabilize the trimer structure, and reduce chromatin-associated PCNA, thereby, selectively inhibiting tumor cell growth, and inducing apoptosis [20].

7)      The characteristics of AgNPs should be given (e.g. what is their size?). Alternatively, reference to the publication where such characteristics can be found, should be given.

We appreciate the comment, we have now also included the main characteristics of the AgNp-G nanoparticles, coupled with the reference in the introduction section (line 60-63), as follows:

Previously, our research group synthesized beta-D-glucose-reduced silver nanoparticles (AgNPs-G) characterized by having an average size of 5.991 nm, quasi-spherical morphology and a −7.48 mV zeta potential value. We demonstrated

We also wrote the reference of the synthesis publication and a summary of such methodology in line 284, so that readers can find more detailed information if wished.

8)      Description of the experimental model is incomplete:

-          route of AgNPs administration is unknown; We have now included the route of administration in section 4.4. Air Bag Model, tumor implantation and treatment administration (line 299-300).

-          AgNPs dose should be calculated an administrated in mg/kg body mass just like doxorubicin; We have added the dose conversion in 1.08 mg/kg .

-          how were tumor dimensions measured?

We added the use of a vernier caliper as our measurement instrument and we delve into the specifics of tumor measurement in section 4.4. Air Bag Model, tumor implantation and treatment administration (line 301-307), as follows:

Tumor volume was measured daily using a vernier caliper until the day of sacrifice (30 days after inoculation of 4T1 cells) and was defined using the following equation [37].

Tumor volume: 4/3 × π × L/2 × W/2 × h/2

Where L corresponds to the longest side, W to the shortest side, and h to the height of the tumor.

The formula referenced for tumor measurement was taken from the following paper: Sápi, J.; Kovács, L.; Drexler, D. A.; Kocsis, P.; Gajári, D.; Sápi, Z. Tumor Volume Estimation and Quasi-Continuous Admin-istration for Most Effective Bevacizumab Therapy. PLOS ONE 2015, 10 (11), e0142190. https://doi.org/10.1371/journal.pone.0142190.

-          how many animals were in each experimental group?

We detailed the experimental design in the methods Line 296-297 “The mice were randomly divided into three experimental groups (n=6)”. 

-          Page 13 line 380: “All experiments were carried out in triplicate” – what does it mean? Three animals per group? Three replicates of each measurement? Whole experiment was repeated three times? Please explain.

This phrase means that whole experiment was repeated three times.

9)      Is it possible that as a result of using B-D-Glucose as a reducing agent, B-D-Glucose is present on the surface of AgNPs and is responsible (at least partially) for the observed effects?

This is an interesting perspective that we also have, we have hypothesized that the presence of glucose in the nanoparticles could promote cell entrance due to the high metabolism of cancer cells, once inside the cell, the silver ions can exert their cytotoxic function. However, our results at the moment cannot support such statement, we know that glucose acts as a reducing agent in the synthesis, but neither the presence of glucose at the surface nor encapsulated has been proven, thereby we opted to report just the premises we can confirm.

10)   Did the authors exclude the presence of pyrogens in AgNPs solution used in the experiments?

We did not test for pyrogens in the AgNPs solution; however, mice were monitored daily for abnormalities, and no clear signs of fever were observed in the mice characteristic of endotoxins. Also no found abnormalities in biochemical parameters related to acute inflammation. Thank you for your recommendations and a LAL test should be considered by us in our next experimental in vitro related to inflammation.

Reviewer 2 Report

Comments and Suggestions for Authors

Comment. 

1. The toxicity of silver NPs was completely studied in vivo/in vitro for both normal and cancer cells. Authors should describe their novelty 

2.  Although many previous reports have been published in the same field, it is not allowed to transfer the findings of silver NPs to clinical trials.  The authors should explain their findings compared to the previous one.

3. In section "2.8. Treatment with AgNPs-G did not generate histomorphological changes in the liver, kidney,  lung, heart, or brain". However, many necrotic stages appeared. 

4. Authors did not think about targeting therapies and how to move silver NPs  directly into cancer locations that could minimize their distribution and prevent their toxicity 

5. Authors must change the title from "β-D-Glucose—Reduced silver nanoparticles remodel the tumor 2 microenvironment in a murine model of triple-negative breast 3 cancer"  to  Evaluation the Toxicity of β-D-Glucose—Reduced silver nanoparticles remodel the tumor 2 microenvironment in a murine model of triple-negative breast 3 cancer".

Comments on the Quality of English Language

 Moderate editing of English language required

Author Response

Reviewer 2

Dear reviewer we would like to thank you for your comments and attentive review, it greatly helped us improve the manuscript and correct mistakes that were overlooked. 

  1. The toxicity of silver NPs was completely studied in vivo/in vitro for both normal and cancer cells. Authors should describe their novelty

The toxicity of silver NPs has indeed been studied previously, however, the main focus the manuscript is the effect of the AgNPs-G on the tumor microenvironment remodeling and the immune response, something that has not been explored in other AgNPs synthesis with β-D-Glucose, and that we consider of great relevance to the field as limited data is available.

  1. Although many previous reports have been published in the same field, it is not allowed to transfer the findings of silver NPs to clinical trials. The authors should explain their findings compared to the previous one.

We evaluated the effect of AgNPs on immunocompetent mice, therefore the previous reports evaluated the effect of AgNPs in immunodeficient mice (References 18, 19). Moreover, the synthesis used is different in each report and we focused on tumor microenvironment. For this is complicated stablished comparative and discussion.   

  1. In section "2.8. Treatment with AgNPs-G did not generate histomorphological changes in the liver, kidney, lung, heart, or brain". However, many necrotic stages appeared.

Thank you for your observation, the line change 179-180 was modify by:

2.8. Treatment with AgNPs-G did generate histomorphological changes in the liver, kidney, lung, heart, or brain.”

And line 260-261 was modify by the phrase:

“…the increase in these hepatic enzymes is indicative of damage in certain organs that correlated with histological data.”

  1. Authors did not think about targeting therapies and how to move silver NPs directly into cancer locations that could minimize their distribution and prevent their toxicity

We have previously found that intraperitoneal administration of the AgNPs-G leads to the accumulation of silver in the tumor and considering that our synthesis produces small-size nanoparticles (average 5.991 nm), we think that such an effect is given by factors such as the Enhanced Permeability and Retention (EPR) effect. Although we have not deepen into the toxicity we also found that it is principally distributed in filtering organs such as gallbladder, kidney and liver where they are probably excreted (Previous research: https://www.frontiersin.org/journals/pharmacology/articles/10.3389/fphar.2023.1332439/full ).

Since the present study was limited to the tumor microenvironment and antitumoral response, the peritumoral route of administration was preferred to focalize the study of the tumor area, based on the results obtained, we can delve into the optimization of the treatment in future research.

  1. Authors must change the title from "β-D-Glucose—Reduced silver nanoparticles remodel the tumor 2 microenvironment in a murine model of triple-negative breast 3 cancer" to Evaluation the Toxicity of β-D-Glucose—Reduced silver nanoparticles remodel the tumor 2 microenvironment in a murine model of triple-negative breast 3 cancer".

We appreciate the comments of the reviewers, and we consider the changes made have improved the quality of the manuscript, however, regarding the title, since the main aim of the study is not to evaluate the toxicity but rather to study the effect of the AgNPs on the tumor microenvironment and immune response we consider that the current title is more fitting.

Round 2

Reviewer 1 Report

Comments and Suggestions for Authors

The manuscript has been improved, but I’m not satisfied with some of the responses. In my opinion, still some improvements are needed before the manuscript can be accepted for publication in IJMS.Below are my comments to the points from my original review:

1) The sentence is still incorrect. To the best of my knowledge, the leading cause of death of women in the US are heart diseases, the second is cancer. Moreover, the leading cause of death from cancer is lung cancer and breast cancer is the second. Please check again, correct and give reference to the latest data.

2) I’m not satisfied with the response. Doxorubicin is well-known drug and two main mechanisms of its action are proposed: (i) intercalation into DNA and disruption of topoisomerase-II-mediated DNA repair and (ii) generation of free radicals. It is well-known that the mechanism of toxicity of silver nanoparticles also involves generation of free radicals and reactive oxygen species. Therefore there is no need to perform clinical trials in dogs to notice similarities between mechanisms of action of doxorubicin and AgNPs and to hypothesize that such similarities are responsible for similar outcomes of the experiments performed with doxorubicin and AgNPs.

3) ok

4) I’m not satisfied with the response. The presented logic behind considering particular p-value as statistically significant or not is still not clear to me.

5) ok

6) ok

7) Ok, but it would be more appropriate to give information concerning AgNPs size, morphology and zeta potential in section 4.3 instead of introduction.

8) The authors responded that the phrase “All experiments were carried out in triplicate” means that whole experiment was repeated three times. If there was 6 mice per experimental group and the experiment was repeated three times, the final results are means from 18 mice per experimental group (6 mice per group x 3 experiments). Correct?

9) I understand, that the presence of glucose at the surface or inside AgNPs has not been confirmed, but the authors should write in the discussion section that there is such possibility.

10). ok

Author Response

REVIEWER 1

The manuscript has been improved, but I’m not satisfied with some of the responses. In my opinion, still some improvements are needed before the manuscript can be accepted for publication in IJMS. Below are my comments to the points from my original review:

We acknowledge your comments and the time taken to review our manuscript.

Your observations are essential to us. In response to this, we comment on the following:

1)The sentence is still incorrect. To the best of my knowledge, the leading cause of death of women in the US are heart diseases, the second is cancer. Moreover, the leading cause of death from cancer is lung cancer and breast cancer is the second. Please check again, correct and give reference to the latest data.

Thank you for your observation, the sentence of line 30-31 was modified by:

Breast cancer (BC) is the most commonly diagnosed neoplasm and the second cause of death in women in the United States [1].

and the reference in line 415 was modified by:

  1. Keys Statistics for Breast Cancer. Available online: cancer.org (accessed on July 26, 2024).

2) I’m not satisfied with the response. Doxorubicin is well-known drug and two main mechanisms of its action are proposed: (i) intercalation into DNA and disruption of topoisomerase-II-mediated DNA repair and (ii) generation of free radicals. It is well-known that the mechanism of toxicity of silver nanoparticles also involves generation of free radicals and reactive oxygen species. Therefore there is no need to perform clinical trials in dogs to notice similarities between mechanisms of action of doxorubicin and AgNPs and to hypothesize that such similarities are responsible for similar outcomes of the experiments performed with doxorubicin and AgNPs.

Thank you for your comment, the following paragraph was added in line 204-209:

The mechanism of action of doxorubicin is based on the damage it causes to cellular DNA and can produce free radicals [20]. These molecular mechanisms also arise in the treatments based on other AgNPs [21]. For this, we can hypothesize that perhaps our nanoparticles have a mechanism of action similar to this chemotherapeutic used as a control. However, it is crucial to perform clinical trials that allow us to determine with certainty that our AgNPs and doxorubicin have similar mechanisms of action.

References:

  1. Goto, S.; Ihara, Y.; Urata, Y.; Izumi, S.; Abe, K.; Koji, T.; Kondo, T. Doxorubicin‐induced DNA intercalation and scavenging by nuclear glutathione S‐transferase π. The FASEB Journal 2001, 15(14), 2702-2714. https://doi.org/10.1096/fj.01-0376com
  2. Butler, K. S.; Peeler, D. J.; Casey, B. J.; Dair, B. J.; Elespuru, R. K. Silver nanoparticles: correlating nanoparticle size and cellular uptake with genotoxicity. Mutagenesis 2015, 30 (4), 577-591. https://doi.org/10.1093/mutage/gev020

3) ok

4) I’m not satisfied with the response. The presented logic behind considering particular p-value as statistically significant or not is still not clear to me.

Thank you for your comment, we modified this for the general format: p ≤ 0.05 (*) and ns (not significant).

5) ok

6) ok

7) Ok, but it would be more appropriate to give information concerning AgNPs size, morphology and zeta potential in section 4.3 instead of introduction.

Thank you for your observation, the phrase in line 61-62 …”characterized by having an average size of 5.991 nm, quasi-spherical morphology, and a −7.48 mV zeta potential value..” was deleted and the phrase:

“These AgNPs were previously characterized and have an average size of 5.991 nm, quasi-spherical morphology, and a −7.48 mV zeta potential value.”

Was added in line 299-301.

8) The authors responded that the phrase “All experiments were carried out in triplicate” means that whole experiment was repeated three times. If there was 6 mice per experimental group and the experiment was repeated three times, the final results are means from 18 mice per experimental group (6 mice per group x 3 experiments). Correct?

Yes, it is correct.

9) I understand, that the presence of glucose at the surface or inside AgNPs has not been confirmed, but the authors should write in the discussion section that there is such possibility.

Thank you for your comment, this phrase was added in lines 200-203:

These mechanisms may be facilitated by incorporating beta-D-glucose in synthesizing our particles, which could be present on their surface, thereby enhancing their interaction with cancer cells. However, the exact location of beta-D-glucose in our particles remains unknown.

 10). ok

Reviewer 2 Report

Comments and Suggestions for Authors

Dear authors 

Silver NPs are so toxic even those prepared by green chemistry.  According to our knowledge, they can not be used for clinical cancer therapies. Thus evaluating their effect on cancer could be more suitable 

The manuscript was revised point by point and can be accepted.

Comments on the Quality of English Language

Moderate editing of English language required

Author Response

REVIEWER 2

Dear authors 

Silver NPs are so toxic even those prepared by green chemistry.  According to our knowledge, they can not be used for clinical cancer therapies. Thus evaluating their effect on cancer could be more suitable.

The manuscript was revised point by point and can be accepted.

We appreciate your comments and the time you took to review our manuscript. Your observations are invaluable to us and will be considered in our future research.

Comments on the Quality of English Language: Moderate editing of English language required

Thank you for your observation. The manuscript has been reviewed and edited by an expert in the English language.
